# PREDICATE-ARGUMENT RELATIONS IN THE HUMAN BRAIN

## ABSTRACT

Humans perceive, understand, and describe the world in terms of relations between things: *Who did what to whom.* Novel stimulus design, neuroimaging (fMRI) data, and machine-learning analysis methods allow assessing the degree to which the human brain represents this information in a way that can be decoded across different subjects and modalities. Analysis of the voxels involved in this decoding demonstrates significant commonality across pairs of subjects and modalities. This suggests a shared neural substrate that supports predicate-argument relations in multiple modalities that is common across different people.

## 1 INTRODUCTION

People perceive, understand, and describe the world in terms of relations between things: *Who did what to whom.* Linguists refer to these as *predicate-argument relations* or *thematic-role assignment.* Predicate-argument relations occur both in language and in vision. One can see whether *John walked and Mary talked* or *John talked and Mary walked* as well as talk about it. That language could be spoken, signed, or written. A two-year-old child can distinguish between *John walked and Mary talked* and *John talked and Mary walked.* Modern deep-learning systems struggle to do this, despite immense amounts of training data. Here, we use neuroimaging (functional magnetic-resonance imaging, fMRI) to study how the adult human brain does this. Our long-term goal is to understand the precise underlying mechanisms so that we can endow computer systems with the same mechanisms. But we are currently very far from that goal. Our shorter term goal is to understand how and where in the brain this computation takes place. While we are getting closer to that goal, it is still out of reach.

Here, we ask three more modest and quantifiable questions:

1. To what extent can we decode predicate-argument relations from fMRI data of human subjects, from both linguistic and nonlinguistic stimuli?
2. To what extent can we decode these across subject? If we can, to what extent does this involve common brain regions?
3. To what extent can we decode these across modality? If we can, to what extent does this involve common brain regions?

If we can decode predicate-argument relations from fMRI data for a particular stimulus modality, this can answer the first question and open the door to asking and answering the next two questions. If we can decode across subject, *i.e.*, by training on one (collection of) subject(s) and testing on another, despite the fact that these subjects have had vastly different life experiences, this can answer the second question and give some evidence that the neural basis of this mechanism is somehow tied to some specific neural hardware or architecture and not just emergent from the training data. If we can decode across modality, *i.e.*, by training on one modality and testing on another, this can answer the third question and give some evidence that the same underlying neural mechanism at the same underlying space-time location of brain processing represents the same decoded information.

To answer these questions, we constructed carefully designed stimuli, each a cross product of a subject (who), a verb (did what), and an object (to whom). These were carefully designed so that they could be rendered both linguistically (as text stimuli) and nonlinguistically (as video stimuli). The set of stimuli was counterbalanced; it contained all possible combinations with equal representation. We conducted a controlled study: the stimuli were carefully constructed not to vary along any other irrelevant axes. A collection of subjects was presented exactly the same set of stimuli, but in

randomized order, while undergoing fMRI recording. By design, the ground-truth predicate-argument relations of each stimulus presentation were known. We trained and evaluated a simple machine-learning model for decoding these relations from the fMRI recordings. This simple decoding model allowed a further analysis of the degree of commonality of voxels supporting the decoding across pairs of subjects and modalities.

Crucial to the design of this experiment was the fact that each stimulus presented two simultaneous subject-verb-object triples. So for example, *Dan* could be *picking up* a *briefcase* while *Scott* could be *putting down* a *chair*. Contrasting this with another stimulus that depicted *Scott picking up* a *chair* while *Dan* was *putting down* a *chair*, allowed us to decode *Who did what to whom*.

## 2 RELATED WORK

Prior work (*e.g.*, Indefrey et al., 2004; Grodzinsky, 2006; Santi & Grodzinsky, 2007; Thompson et al., 2007; Grodzinsky & Santi, 2008; Fazio et al., 2009; Willems & Hagoort, 2009; Rogalsky & Hickok, 2011; Brennan et al., 2012; Fedorenko et al., 2012; Frankland & Greene, 2015; Zaccarella & Friederici, 2015; de Heer et al., 2017; Hale et al., 2018; Bhattasali et al., 2019; Deniz et al., 2019; Fedorenko et al., 2020; Matchin & Hickok, 2020; Pylkkänen, 2020; Reddy & Wehbe, 2021; Matchin et al., 2022; Parrish & Pylkkänen, 2022; Toneva et al., 2022; Zhang et al., 2022; Liu et al., 2023; Bulut, 2023; Tang et al., 2023) has studied syntactic and linguistic processing in the brain. Our work here differs from this prior work in a number of key ways:

- Most prior work studies sentence processing through reading, speech comprehension and production, and sign-language comprehension, all of which explicitly encode sentences. In contrast, half of our stimuli are short video clips of human activity that only implicitly encode sentences.
- Most prior work attempts to localize processing, not decode. While this can address the question of the degree to which the locus of processing is similar across subjects and/or modalities, it inherently can't address the question of whether the nature of processing is similar. We address this question by decoding rather than localizing.
- Most prior work uses stimuli of a single modality. We use stimuli of two very different modalities, one linguistic, one nonlinguistic, and decode both within and across these modalities.
- We know of no prior work that can decode across subjects and/or across modalities.
- Most prior work attempts to understand sentence processing in general, and syntax in particular. In contrast, we focus on predicate-argument relations. While these are manifest in language, since they appear to also manifest in vision, the brain processing underlying predicate-argument relations might not be particular to sentence processing, syntax, and even language, but rather might be a more general property of how the brain encodes representations of the world.
- Prior work (Tang et al., 2023) on decoding sentences does so by producing a string of words with a language model conditioned on fMRI but does not explicitly decode predicate-argument relations.

We go far beyond prior work in that we don't seek just to decode individual noun/object concepts like *Dan* and *chair*, or even individual verb/activity concepts like *pick up*, by additionally decoding concept **pairs**, like *Dan pick up* and *pick up chair*, that constitute (subject-verb, verb-object, and subject-object) **predicate-argument relations**. We further do this for both **linguistic** and **nonlinguistic** stimuli.

## 3 METHOD

The supplementary material contains all code used to conduct the experiments herein. We will release the stimuli and data collected upon publication. (They don't fit in the 100MB limit.) Some material has been omitted as it is difficult to anonymize. It will be added upon publication.

### 3.1 STIMULI

Each stimulus depicted a pair of simple compositional concepts, each corresponding to a subject-verb-object (SVO) triple, where the *subject*, *verb*, and *object* were optional.

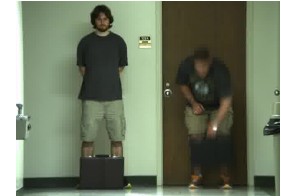

$$\left( \left\{ \begin{array}{l} \textit{Dan} \\ \textit{Scott} \\ \textbf{no subject} \end{array} \right\} \times \left\{ \begin{array}{l} \textit{pick up} \\ \textit{put down} \\ \textbf{no verb} \end{array} \right\} \times \left\{ \begin{array}{l} \textit{briefcase} \\ \textit{chair} \\ \textbf{no object} \end{array} \right\} \right)^2$$

Figure 1: A frame from a sample video stimulus depicting *Dan and a briefcase are on the left* and *Scott picked up a briefcase on the right*.

One triple corresponded to something on the left, the other to something on the right. The two activities could optionally exhibit the same verb or different verbs. As we had two briefcases and two chairs, both activities could optionally involve the same object, or different objects. However, only one could optionally involve *Dan* and only one could optionally involve *Scott*. An activity that included a verb necessarily included both a subject and an object. But a subject and/or object could appear without a verb, subject to the constraint that each stimulus depicted at least one verb. The combinatorics are such that there are 128 possible stimuli.

Stimuli were presented in two *modalities*: *video* and *text*. Four short video clips (4 s, 40 frames, 640×480 at 10 fps) were filmed depicting each of the 128 possible stimuli (Fig. 1). All clips were filmed in the same location at the end of a hallway with the same camera viewpoint and no other variation in the background. We used two identical folding chairs and two nearly identical briefcases, differing only slightly in color. Dan and Scott wore nearly identical clothing, which was the same for all clips. This increased the likelihood that subjects could focus on the relevant concept. The triples were mechanically rendered as grammatical sentences by converting verbs to past tense and introducing a minimal number of the words *a*, *and*, *is*, *are*, *nothing happens*, *on the left*, and *on the right*. Text stimuli were machine generated in the form of pairs of sentences, rendered in two adjacent lines in random fonts, point sizes, and positions in the field of view (uniformly distributed, sampled with replacement). This increased the likelihood that we would decode the concept semantics, not the visual characteristics of the rendering.

## 3.2 DATA COLLECTION

All protocols, experiments, and analyses were carried out with approval of the Institutional Review Board at REDACTED University (protocol REDACTED). Fourteen adult subjects, some students, were recruited from the general population of REDACTED. Twelve (six male, six female, ages 20 to 36, mean age 25;1) completed the study and their data was included. Two did not and their data was not included.

Subjects were screened for safety, eye/hand dominance, and fluency. Only right-handed subjects (as some left-handed people have language areas in the opposite brain hemisphere) and only those with native English fluency from early childhood (since our linguistic stimuli were in English) were recruited. Informed consent was obtained from all subjects. Subjects were paid $70 per four-hour session. Subjects were shown a slide show with instructions. They were told the overall design of the experiment and the structure of the stimuli, were shown images of *Dan* and *Scott*, told to think of sentences that depicted the video clips, told to read sentences depicted as text, but to just think of these sentences, not to read them out loud (to decode perception, not motor control). There was no other task. Each participated in two sessions, each session with eight functional (EPI) runs. One session acquired an anatomical (3D FSPGR) at the end. (This scan was not used in the current study.) A few subjects completed each session without exiting the scanner. Most, however, exited the scanner at various points during the set of runs, which required cross-session registration.

Imaging was performed at the Engineering MRI Facility at REDACTED University using a GE Discovery MR750 3T scanner (Waukesha, Wisconsin) with a Nova Medical (Wilmington, Massachusetts) 32-channel brain array to collect whole-brain volumes via a gradient-echo EPI sequence with 2,000 ms TR, 200 mm×200 mm FOV, 77° flip angle, and 26 ms TE. Thirty-five axial slices were acquired with a 3.0 mm slice thickness using a 64×64 acquisition matrix resulting in 3.125 mm×3.125 mm×3.0 mm voxels. The anatomical had 168 sagittal 256×256 slices and a 240 mm×240 mm FOV resulting in 0.9375 mm×0.9375 mm×1.0 mm voxels. Runs were separated by several minutes, during which no stimuli were presented, no data was gathered, and subjects engaged in unrelated conversation with the experimenters. Each run contained the same number of stimulus presentations. Each run

began with 4 TRs of fixation, contained 64 stimulus presentations, and ended with 10 TRs of fixation. Each stimulus presentation lasted 2 TRs and was followed by at least 1 TR of fixation. Additional jitter fixation, in integral multiples of TRs, was randomly inserted after each stimulus presentation, constrained so that the runs for each dataset had the same duration (254 TRs).

Runs were counterbalanced. Each run had exactly 32 presentations of video stimuli and 32 presentations of text stimuli, randomly interleaved for a total of 1,024 stimulus presentations per subject. Each subject saw each of the $4\times128$ video clips exactly once and saw each of the 128 text sentences exactly 4 times, sampled uniformly without replacement. Jitter and choice of stimulus and stimulus modality in each presentation slot varied randomly across subject.

## 3.3 PREPROCESSING

The raw functional scans had 143,360 voxels per volume. Uniform preprocessing was performed on all runs for all subjects. Whole-brain scans were processed using AFNI (Cox, 1996) and PyMVPA (Hanke et al., 2009) to drop the first two TRs of each run, motion correct each run, align all volumes for all runs for all subjects to a specific reference volume for a specific subject (`3dWarpDrive -affice_generate -cubic`), skull-strip all volumes for all runs for all subjects with a common brain mask computed over all volumes for all runs for all subjects (`3Automask` followed by ORing), and detrend each run (`poly_detrend polyord=4`). This yielded volumes with 60,732 voxels. Performing alignment and skull stripping in common for all runs of all subjects facilitated subsequent cross-subject analyses as all scans had the same number of voxels after skull stripping and these voxels nominally were in alignment. All analyses were performed on this same preprocessed data.

Voxels within a run were z-scored, subtracting the mean value of that voxel for the run and dividing by its variance. Z-scoring was done using only the volumes associated with stimulus presentations. While each stimulus was presented for 2 TRs, all analyses, and thus z-scoring, employed a single volume for each stimulus presentation, the third TR after the onset of stimulus presentation, to account for the hemodynamic response function.

## 3.4 CONCEPTS

Various analyses were performed to detect a variety of concepts in the recorded brain data. Attempts were made to detect two kinds of concepts: *single concept*, namely those corresponding to the individual words *Dan*, *Scott*, *pick up*, *put down*, *briefcase*, and *chair*, and *concept pair*, namely those corresponding to pairs of words. The latter could be subject-verb (SV), like *Dan pick up*, verb-object (VO), like *pick up briefcase*, or subject-object (SO), like *Dan briefcase*. Trials were labeled *present* if they depicted a target concept and *absent* otherwise. A concept pair like *Dan pick up* was considered present if Dan picked something up. It was considered absent, even if both *Dan* and *pick up* were present, if Dan was not picking something up, *i.e.*, if Dan were doing something else and someone else picked something up. Note that a trial could be labeled as present for one concept but absent for another. And a given trial could have more than one concept present. Thus which trials were considered as present *vs.* absent varied depending on which concept was being detected. This present-absent annotation was provided as ground-truth labeling to training and used during test for evaluation. The annotation was available by design because stimuli were filmed or rendered from text to depict a known annotation.

## 3.5 SUBSETTING

The entire dataset consists of 1,024 trials for each of 12 subjects, organized into 16 runs per subject, each run with 64 trials. Some analyses were performed *within modality*. These considered only those trials of a specific modality, namely video or text, both for training and test. Other analyses were performed *cross modally*. These considered only those trials of one modality for training and trials of the other modality for test. For example, video→text analyses considered video trials for training and text trials for test. In plots of cross-modal analyses below "video" denotes text→video while "text" denotes video→text.

All analyses were performed on all trials of the requisite modality. Subsetting preserved the trial order and assignment of trial to subject and run. A total of 6 single concepts and 12 concept pairs (4 each of SV, VO, and SO) were analyzed, for each modality.

### 3.6 SPLITS

Three kinds of analyses were performed, to answer each of the three questions.

*Pooled subject* analyses were performed independently on each modality with 8-fold leave-one-partition-out round-robin cross validation. After subsetting, the runs for each subject were concatenated in order. The trials for all 12 subjects were interleaved (*i.e.*, a 12-way extension of the concept of the perfect shuffle). These were then divided into 8 equal-sized contiguous partitions. For a given fold, 7 partitions were taken as the training set, and 1 partition was taken as the test set. A distinct model was trained and tested for each combination of concept, fold, and modality ($18\times8\times2{=}288$).

*Cross subject* analyses were performed independently on each subject and modality. After subsetting, the runs for each subject were concatenated in order. For each of the 12 target subjects, the trials of the other 11 source subjects were interleaved and taken as the training set. The trials of the target subject were taken as the test set. A distinct model was trained and tested for each combination of concept, target subject, and modality ($18\times12\times2{=}432$).

*Pooled subject cross modal* analyses were performed both for video→text and text→video. After subsetting, the runs for each subject were concatenated in order. The trials for all 12 subjects were interleaved. All trials of the source modality were taken as the training set and all trials of the target modality were taken as the test set. A distinct model was trained and tested for each combination of concept and direction of modality transfer ($18\times2{=}36$).

All analyses except pooled subject cross modal were within modality. With this design, for each model trained and tested, the training and test sets were disjoint. Further, each trial under consideration for each analysis of a concept and modality appeared in exactly one test set for that analysis.

### 3.7 SUPERTRIALS

Neuroimaging data can be noisy. Single-trial training and testing can be hampered by this. Prior work has investigated training and testing EEG classifiers on supertrials or ERPs that are obtained by averaging across multiple trials independently per channel and per time point (Isik et al., 2014; Cichy et al., 2016; Greene & Hansen, 2020; Zheng et al., 2020; Bharadwaj et al., 2023). This can lead to improved classification accuracy by increasing the signal-to-noise ratio.

Here, we apply supertrials to fMRI data. After preprocessing, subsetting, and splitting the trials to be considered into disjoint training and test sets, supertrials were formed independently on the training and test sets. To form supertrials, the data was partitioned into present and absent trials based on the concept under consideration. Present and absent supertrials were formed independently from present and absent trials respectively. Each present (absent) supertrial was formed by averaging the preprocessed data for $K$ distinct present (absent) trials, independently for each voxel. The first supertrial was formed by averaging the first $K$ trials, the second supertrial was formed by averaging trials 1 to $K + 1$, *etc.* wrapping around the trials under consideration. This method of forming supertrials yields the same number of supertrials from a dataset as the number of trials in that dataset, where each supertrial is formed from a distinct set of trials and each trial is used in the same number of supertrials. Note that since the trials are independent, and forming supertrials corresponds to multiplying the trial matrix by another non-rank-deficient matrix, this does *not* reduce the rank of dataset and serves simply to reduce the variance. Again, we stress that this was done independently for the training and test sets to avoid contamination. Further, the reason we interleaved data from different subjects was to form supertrials from a mixture of subjects. $K = 32$ for all analyses here. All subsequent analysis was performed on supertrials.

### 3.8 MODELS

We used a single-layer perceptron (SLP) as the model. The input was a single 60,732-voxel brain volume from a supertrial. The output was a single judgment produced by a single FC layer followed by a sigmoid. Training was performed with MSE loss (taking present as 1 and absent as 0) and SGD with default PyTorch parameters for 2,000 epochs with a batch size of 30 and initial learning rate of 0.01. No shuffling was performed. A learning rate scheduler performed a single change at 1,000 epochs with $\gamma = 0.1$. After training, the model output was taken as the confidence of a prediction for each supertrial in the test set.

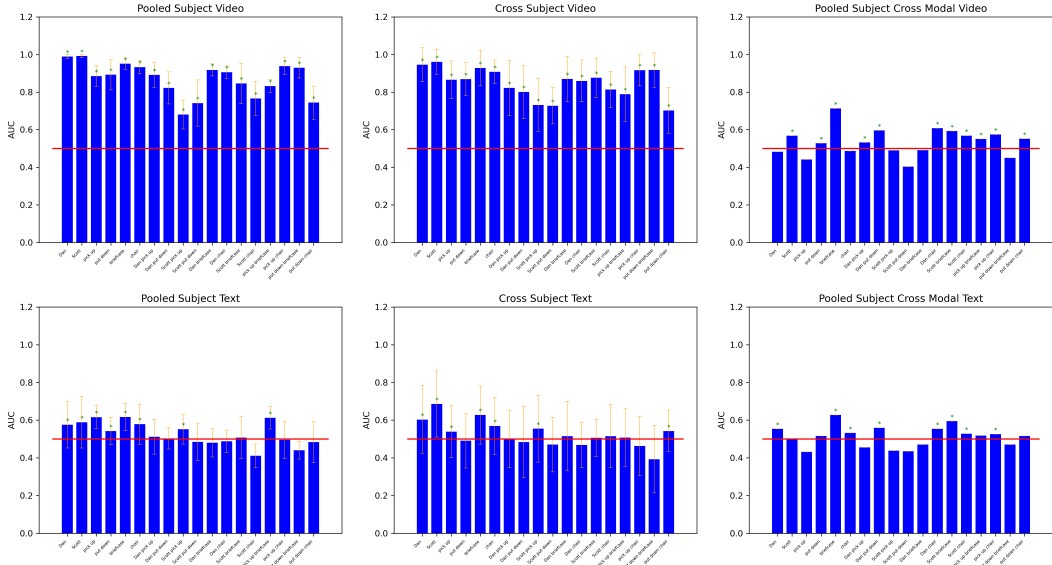

Figure 2: Mean AUC values. For pooled subject, averaged over fold. For cross subject, averaged over subject. For pooled subject cross modal, there is only a single model for each concept and modality, thus no averaging was done. Error bars on pooled subject and cross subject denote one standard deviation from the mean. The red lines denote chance. Stars denote statistical significant above chance ($p < 0.005$).

Such a simple model was used due to the paucity of data. We only have 1,024 trials for 12 subjects, 512 trials for each modality. The brain volume, however, has 60,732 voxels. Thus our model has 60,732 parameters, one per voxel. With so few samples of such large dimension, even such a simple model will overfit. Any more complicated model will overfit even more. This is the nature of neuroimaging. Data is expensive. Scanner time is $600/hr. Adding in subject payments and salary of primary and secondary scanner operators, each data point costs over $6. Moreover, data is tedious to obtain. Obtaining 1,024 data points takes more than a full business day and requires 3 people. This enterprise is inherently different from other machine-learning endeavors where data can be crowdsourced or downloaded from the web. The data collected here is specific to our stimulus design. No other known dataset is suited to analyze the questions we ask here.

## 4 RESULTS

### 4.1 ROC CURVES AND AUC

For each combination of analysis, concept, optionally subject, optionally fold, and modality or direction of modality transfer, depending on analysis, we have a detection confidence for each test trial. Given this, and a specified threshold, we take all detection confidences above the threshold to be predicted presence and below the threshold to be predicted absence. Comparing against ground-truth annotation, and accumulating across all trials yields a confusion matrix for each combination of threshold, analysis, concept, optionally subject, optionally fold, and modality or direction of modality transfer. For each confusion matrix, we compute a point on a receiver operating characteristics (ROC) curve, whose $x$ coordinate is the false positive rate and whose $y$ coordinate is the true positive rate. By varying the threshold, we then produce a curve for each combination of analysis, concept, optionally subject, optionally fold, and modality or direction of modality transfer, depending on analysis. We do this by sorting the detection confidences and selecting the midpoints between adjacent confidences as varying threshold values. We then compute the area under each ROC curve (AUC) via trapezoidal integration. AUC values were averaged over subject and/or fold, and their variance computed. Fig. 2 depicts these graphically.

## 4.2 STATISTICAL SIGNIFICANCE

We assess statistical significance ($p < 0.005$, starred values in the plots) of the mean AUC values against the null hypothesis

$H_0$:  The mean AUC computed on detection confidences sampled from a uniform distribution in $[0, 1]$ against ground truth is greater than or equal to the AUC computed on the detection confidences produced by the model against the same ground truth.

by sampling (16,000 samples).

## 4.3 COMMON VOXELS

Since we use SLPs as detectors, a single weight controls the influence of each voxel on detector output. To assess the degree of commonality of voxels that support cross-subject and cross-modal processing of predicate-argument relations, we compute a rank ordering of the absolute values of the weights for each trained detector. We rank order absolute values of weights since both strongly negative and strongly positive weights influence detector outcome. We compare two rank orderings in two different ways: Kendall correlation (Kendall, 1938) and a novel metric discussed below.

Kendall correlation compares two rank orderings. In our case, the quantities being ranked are voxels that have 3D positions. Due to inaccuracies in alignment, we wish to measure whether voxels of similar importance are spatially close, not necessarily identical. We have four desiderata for a metric:

1. We wish to weight high-rank voxels higher than low-rank voxels. Intuitively, high-rank voxels presumably carry more of the requisite information than low-rank voxels and we care more whether they are the same across subject/modality.

2. We wish to penalize larger differences in rank disproportionately more than smaller differences. Intuitively, we don't care whether one subject/modality has a specific voxel rank 2 and another has it rank 3. We do care if one has it rank 2 and another has it rank 1000.

3. The 3D spatial position of voxels matters. We would expect that a spatially adjacent cluster of voxels of high rank in one subject/modality would also exhibit high rank in another subject/modality. We don't care about small rank difference within the cluster. But we do care if, say, voxels rank 1–100 are in two completely different locations in two different subjects/modalities.

4. We would like some tolerance to misalignment. Due to minor misalignment the rank 1 voxel might be in two different indices in two different subjects yet be spatially very close (*i.e.*, neighboring voxels).

In attempt to satisfy these desiderata, we compute the following metric which we call Weighted Unsquared Chamfer Distance (WUCD). Let $\rho_1$ and $\rho_2$ be two rank orderings we wish to compare and $X$ be a map from voxel indices to 3D positions. We take a moving window of $k$ voxels starting at rank $i$ in two voxels rankings we wish to compare. We then compute a variant of the Chamfer distance (using Euclidean distance rather than its square) on the 3D voxel positions in each window. We then take a weighted average over different windows with different $i$, weighting smaller $i$ higher than larger $i$.

$$d_C(X, Y) = \frac{1}{|X|} \sum_{x \in X} \inf_{y \in Y} \|x - y\|_2 + \frac{1}{|Y|} \sum_{y \in Y} \inf_{x \in X} \|x - y\|_2 \qquad (1)$$

$$d(\rho_1, \rho_2) = \sum_{i=1}^{N-k+1} a_i d_C(X[\rho_1[i : i + k]], X[\rho_2[i : i + k]]) \qquad (2)$$

We take the window size $k = 100$, the number of windows $N = 500$, and the weights $a_i$ to decay steeply exponentially.

We then compute the mean Kendall correlation and WUCD between all pairs of subjects, for both video and text, averaged over all 12 concept-pair analyses. We also compute the Kendall correlation

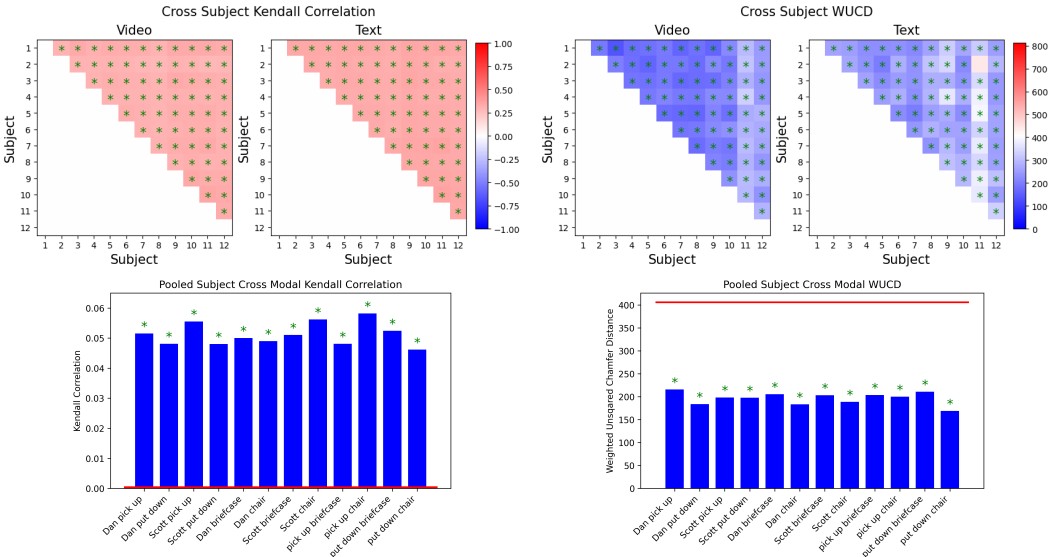

Figure 3: Mean Kendall correlation (top left) and WUCD (top right) between pairs of subjects for cross-subject analyses of concept pairs, averaged over all concept pairs, from video and text stimuli. Kendall correlation (bottom left) and WUCD (bottom right) between pooled-subject-cross-modal for video and text. The red lines denote chance. Stars denote statistical significant above chance ($p < 0.005$).

and WUCD between the pooled-subject-cross-modal analyses for video and text. Fig. 3 depicts these values graphically. Note that higher Kendall correlation and lower WUCD indicate greater commonality of voxels that influence detector outcome.

We assess statistical significance ($p < 0.005$, starred values in the plots) of the (mean) Kendall correlation and WUCD against the null hypothesis

$H_0$: The (mean) Kendall correlation (WUCD) computed on pairs of uniformly sampled random voxel rankings is greater (less) than or equal to the (mean) Kendall correlation (WUCD) computed on the actual pair of voxel rankings.

by sampling (5,000 samples).

## 5 DISCUSSION

As can be seen from Fig. 2, the pooled-subject analyses are considerably above chance for all concepts for video. This strongly answers the first question in the affirmative for video. The pooled-subject analyses for all single-concept text are also above chance. But only two of the pooled-subject analyses for concept-pair text are above chance. This leaves the first question for text unanswered. It is unclear why it is possible to decode single concepts but not concept pairs.

The cross-subject analyses are also all considerably above chance for all concepts for video. This strongly answers the second question in the affirmative for video. The cross-subject analyses for five of the six single-concept text are also above chance. But only two of the cross-subject analyses for concept-pair text are above chance. This leaves the second question for text unanswered.

Eleven out of eighteen of the text→video and eight out of eighteen of the video→text pooled-subject-cross-modal are above chance. These include a mix of single concepts and concept pairs. This provides weak evidence for answering the third question in the affirmative.

Fig. 3 further strengthens the above affirmative results and serves to resolve all of the unanswered questions in the affirmative. All but one of the cross-subject analyses and all of the pooled-subject-cross-modal analyses have statistically significant above chance Kendall correlation and WUCD.

Since we select $N = 500$ and $k = 100$ voxels, that corresponds to less that 1% of the brain volume, together with a steeply exponential $a_i$, this suggests that a fairly small cluster of voxels accounts for predicate-argument relations and is shared across both subject and modality.

Our work here has important implications for future approaches to AI, computer vision, and natural-language processing. The fact that we can decode predicate-argument relations across subject and modality, and that such decoding uses a small common set of voxels shared across subject and modality, suggests that human brain processing operates in a vastly different fashion from large language models and foundation vision-language models, or more generally any deep-learning method or even any machine-learning method. One could imagine attempting 'simulated fMRI' to decode predicate-argument relations from the internal activation state of a neural network. One could further imagine attempting this across two different instances of the same network trained on different training data, to simulate cross-subject analyses, or across two different networks, say one an LLM and the other a foundation vision model, to simulate cross-modal analyses. And one could even further imagine performing the Kendall correlation and WUCD analysis between pairs of rank orderings of internal neural-network activations. One would conjecture that such decoding would not work and the corresponding Kendall correlation and WUCD would not support the conclusion of a common small set of activations influencing the operation of the neural network. Yet that is exactly what we are doing here with human neuroimaging data. Different subjects have had vastly different life experiences, and saw and heard vastly different things. Yet we can decode across subject reliably and across modality, albeit more weakly. And this is supported by a small common set of voxels representing less than 1% of the brain volume.

## 6    CONCLUSION

We have presented a novel framework to investigate predicate-argument relations in the human brain. Novel experiment design includes stimuli that present two simultaneous SVO triples, where both verbs can be the same or different, and both objects can be the same or different. This allows attempting to decode the correspondence between subjects, verbs, and objects. Novel experiment design includes exactly the same stimuli in two modalities: video and text. This allows attempting cross-modal decoding. Moreover, success appears to hinge on inclusion of a visual modality as this appears to have a representation that can more easily be decoded with the spatio-temporal resolution of current fMRI technology. We collected a large novel neuroimaging dataset based on this design. Formulating the machine-learning analysis as a detection problem, analyzed with ROC curves and AUC measures, together with supertrials, supports decoding that was not possible without supertrials and by using a more traditional classifier approach. Finally, novel methods for assessing the commonality of voxels that support decoding of brain activity across subject and modality, suggest a small neural substrate underlying brain processing of predicate-argument relations that is common across subject and modality.

AUTHOR CONTRIBUTIONS

REDACTED

ACKNOWLEDGMENTS

REDACTED

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
