# OpenReview forum: "Predicate-Argument Relations in the Human Brain"
_ICLR.cc/2024/Conference — ICLR 2024 Conference Withdrawn Submission_

### Official Review · Reviewer_ZsMm · 2023-11-02

**Soundness:** 1 poor
**Presentation:** 2 fair
**Contribution:** 1 poor
**Rating:** 3
**Confidence:** 3

**Summary:**

The manuscript describes the data collection process for predicate-argument relations. The dataset should allow brain decoding to correspond between subjects, verbs, and objects. They developed a single-layer perception model to perform brain decoding and used it to find common voxels that support cross-subject and cross-modal processing of predicate-argument relations.

**Strengths:**

- Interesting dataset

**Weaknesses:**

- The manuscript has not considered other baselines from brain decoding literature (e.g., Searchlight (Etzel et al., 2013)). Since the significance and correctness of the proposed methodology is not clear.
- The manuscript does not use a hold-out set; it only uses training and validation sets. It would be best to have an additional split since the validation set should be used only for checkpointing and hyperparameter search, not for reporting the final performance. Hence, the better strategy is to perform a nested cross-validation strategy.
- Ablation of hyperparameters is missing.
- "No other known dataset is suited to analyze the questions we ask here." You might find it helpful to use pretraining methods on other resting-state or task-data fMRI datasets and fine-tune these models to your dataset.
- There is no visualization for the common voxels that were detected and no discussion about which regions were activated and how they differ in different settings.

Etzel, Joset A., Jeffrey M. Zacks, and Todd S. Braver. "Searchlight analysis: promise, pitfalls, and potential." Neuroimage 78 (2013): 261-269.

**Questions:**

- It is unclear why you would use MSE for 0 and 1 targets.
- Have you ensured that your model's probabilities are well calibrated? This is an essential step before computing the ROC.
- Given the description in Sections 4.2 and 4.3, have you run corrections for multiple comparisons? Also report the effect sizes.
- The font in Figure 2 is too small.

---

### Official Review · Reviewer_HNdL · 2023-11-03

**Soundness:** 2 fair
**Presentation:** 2 fair
**Contribution:** 2 fair
**Rating:** 3
**Confidence:** 4

**Summary:**

This work aims to decode predictate-argument structure from fMRI recordings of participants viewing text and video stimuli. The data was specifically recorded for this study and the authors say that they will release it upon publication. Binary linear decoders are trained to predict whether a specific concept or a concept pair from the set {(subject, verb); (verb, object), (subject,object)} is present or absent in a specific brain recording. The results show that the concepts and concept pairs can be decoded from fMRI corresponding to videos and across participants, and concepts but not concept pairs can be decoded from text.

**Strengths:**

- Investigate two sensory modalities in the brain
- Will release data and code upon publication

**Weaknesses:**

1. Severe lack of clarity in many parts of the manuscript, see specific questions below. This really hampers understanding the contributions of the work.

2. Writing and structure can be much improved. Some parts of the manuscript are too brief and lack motivation (e.g. the stimulus design, motivation for looking across multiple modalities, background on what should be expected for the visual modality, what the cited related work is actually doing instead of just listing the references). Other parts are too detailed for the main paper of a submission to a machine learning venue (a whole page is spent on data collection and preprocessing). The writing also comes off at times as a bit patronizing but also too informal. A good example is this excerpt “With so few samples of such large dimension, even such a simple model will overfit. Any more complicated model will overfit even more. This is the nature of neuroimaging. Data is expensive. Scanner time is $600/hr. Adding in subject payments and salary of primary and secondary scanner operators, each data point costs over $6. Moreover, data is tedious to obtain.”

3. It’s not clear to me that an ML venue is the best place for this submission. There is no innovation on the methodology, and the results are entirely neuroscience-focused, so it seems that a neuroscience audience will be better able to give feedback and appreciate this work.

**Questions:**

Q1. In the current analysis setting, "Scott pick up" would be considered the same as "pick up Scott" but those two have different predicate argument structures. I would like the authors to comment on how their work studies predicate-argument structure and not just concept co-occurrence.

Q2. Stimulus design:

a. What is the motivation for showing a pair of concepts in one stimulus?

b. Can you further explain how showing the text stimuli in “random fonts, point sizes, and positions in the field of view” is increasing the likelihood that you decode the concept semantics and not visual characteristics? If I understand correctly, each text stimulus is then very likely to have a different combination of font, size, and position, which actually makes it easier to decode if the decoder depends on visual properties.

c. For the stimuli that did not have a subject (e.g. pick up briefcase), how was the video created?

Q3. Data splitting:

a. What “subsetting” was actually done? It seems that Section 3.5 is aimed to explain something but it’s not coming across. Can the authors explain in simple language how what data was trained on and tested on to answer each of the questions?

b. The way the supertrials are implemented seems quite unfair for binary classification. The “present” trials all have a specific concept in common, so averaging over them can reinforce this concept. The “absent” trials don’t necessarily have anything in common, other than not having a particular concept. So averaging over them can destroy important semantic information.

Q4. Analysis choices: please discuss your motivation for the following choices and how you would expect changes in those choices would affect the results:

a. binary classification vs multi class classification

b. the grouping intro super trials

c. the hold out strategy

---

### Official Review · Reviewer_B2cQ · 2023-11-06

**Soundness:** 4 excellent
**Presentation:** 3 good
**Contribution:** 4 excellent
**Rating:** 8
**Confidence:** 4

**Summary:**

The paper investigates the decodability of predicate-argument relations from fMRI data during exposure to both linguistic and nonlinguistic stimuli, seeking to determine the consistency of this decoding both inter-subjectively and across different modalities. Employing innovative stimulus design combined with advanced machine learning techniques on fMRI data, the authors reveal notable commonalities in the brain regions involved in processing these relations among different individuals and modalities. The results indicate a universal neural basis for understanding predicate-argument relations, pointing towards a shared cognitive processing mechanism within the human brain.

**Strengths:**

1. Innovative Research Design: The paper introduces a novel experimental setup, complete with a unique dataset and an open-source baseline model, which marks a significant departure from traditional studies in the field. This pioneering approach not only advances the research on neural representations of language but also provides a solid foundation for future studies to build upon, demonstrating a clear understanding of the need for open science and replicability in neuroimaging research.
2. Exploration of Neural Decoding for Language Processing: The study ambitiously tackles the challenge of decoding the neural correlates of subject-verb-object correspondences from fMRI data, which is a substantial contribution to our understanding of how the brain processes language. The authors' attempt to unravel this complex aspect of predicate-argument relations using neuroimaging data provides valuable insights into the intersection of neuroscience, linguistics, and artificial intelligence.
3. Rigorous Statistical Analysis: The paper doesn't merely present observational findings but backs them with detailed statistical analyses to test the robustness of the results. This statistical rigor ensures that the conclusions drawn from the study are reliable and contribute to the field with evidence-based claims.

**Weaknesses:**

Limited Clarity in Model Description: While the paper presents innovative findings, there is a notable weakness in the model description. The absence of equations, which are typically crucial for conveying complex models with precision, results in a lack of clarity.

**Questions:**

1. How are these hyper-parameters selected in Section 3.8?
2. It appears there is a typo in the sentence "six male, six female, ages 20 to 36, mean age **25;1**".
3. Is the data a time series or does it only contain one frame for each trial in the dataset?
4. In Section 4.3, would it be possible to visualize the common voxels found in a brain template?

**Details Of Ethics Concerns:**

The paper will release a dataset.

---

### Official Review · Reviewer_L5si · 2023-11-08

**Soundness:** 2 fair
**Presentation:** 3 good
**Contribution:** 2 fair
**Rating:** 3
**Confidence:** 5

**Summary:**

In this paper, the authors investigate the neural basis of predicate-argument relations in the human brain through a neuroimaging study. The authors designed novel stimuli in both linguistic and non-linguistic modalities (video and text) depicting two simultaneous subject-verb-object (SVO) triples. The same 128 stimuli in video and text forms were presented to 12 subjects to collect fMRI data. A simple single-layer perceptron was used for decoding. Experimental results show above-chance decoding within and across modalities for video stimuli, while text stimuli decoding was weaker. The authors suggest that the human brain has a shared neural substrate for understanding relations between entities, consistent across different people and types of stimuli.

**Strengths:**

The paper is written clearly and well-structured, and the experiment design is intuitive and relatively easy to follow.

The novel stimulus design, which crosses subjects, verbs, and objects in videos and text, boosts the cross-modal and cross-subject decoding framework of predicate-argument relations.

**Weaknesses:**

The collected fMRI dataset is relatively small, only 12 subjects were used, which may affect decoding accuracy and statistical power. May consider EEG to lower the scanner cost.

Only a simple single-layer perceptron model was used for decoding.
Lack of comparative analysis with other decoding methods such as PCA or more advanced transformer-based autoencoder.

The investigation of text stimuli decoding is limited. A more comprehensive exploration could offer insights into the discrepancy between video and text stimuli and its implications for the understanding of brain processing, and offer potential avenues for enhancing AI to mimic human cognitive mechanisms.

**Questions:**

I do not have additional questions at this stage.